# Glutaredoxin regulation of primary root growth is associated with early drought stress tolerance in pearl millet

Carla de la Fuente[1†], Alexandre Grondin[1,2,3†], Bassirou Sine[2,3], Marilyne Debieu[1†], Christophe Belin[4], Amir Hajjarpoor[1], Jonathan A Atkinson[5], Sixtine Passot[1], Marine Salson[1], Julie Orjuela[1], Christine Tranchant-Dubreuil[1], Jean-Rémy Brossier[1], Maxime Steffen[1], Charlotte Morgado[1], Hang Ngan Dinh[1], Bipin K Pandey[5], Julie Darmau[1], Antony Champion[1], Anne-Sophie Petitot[1], Celia Barrachina[6], Marine Pratlong[6], Thibault Mounier[7], Princia Nakombo-Gbassault[1], Pascal Gantet[1], Prakash Gangashetty[8], Yann Guedon[9], Vincent Vadez[1,2,3], Jean-Philippe Reichheld[10], Malcolm J Bennett[5], Ndjido Ardo Kane[2,3], Soazig Guyomarc'h[1], Darren M Wells[5], Yves Vigouroux[1*‡], Laurent Laplaze[1,2*‡]

[1]DIADE, Université de Montpellier, IRD, CIRAD, Montpellier, France; [2]LMI LAPSE, Dakar, Senegal; [3]CERAAS, ISRA, Thies, Senegal; [4]LGDP, Université de Perpignan, Perpignan, France; [5]School of Biosciences, University of Nottingham, Sutton Bonington, United Kingdom; [6]Montpellier GenomiX, Montpellier, France; [7]Be More Specific, Montpellier, France; [8]ICRISAT, Hyderabad, India; [9]UMR AGAP Institut, Univ Montpellier, CIRAD, INRAE, Institut Agro, Montpellier, France; [10]LGDP, CNRS, Perpignan, France

*For correspondence:
yves.vigouroux@ird.fr (YV);
laurent.laplaze@ird.fr (LL)

[†]These authors contributed
equally to this work
[‡]These authors also contributed
equally to this work

**Abstract** Seedling root traits impact plant establishment under challenging environments. Pearl millet is one of the most heat and drought tolerant cereal crops that provides a vital food source across the sub-Saharan Sahel region. Pearl millet's early root system features a single fast-growing primary root which we hypothesize is an adaptation to the Sahelian climate. Using crop modeling, we demonstrate that early drought stress is an important constraint in agrosystems in the Sahel where pearl millet was domesticated. Furthermore, we show that increased pearl millet primary root growth is correlated with increased early water stress tolerance in field conditions. Genetics including genome-wide association study and quantitative trait loci (QTL) approaches identify genomic regions controlling this key root trait. Combining gene expression data, re-sequencing and re-annotation of one of these genomic regions identified a glutaredoxin-encoding gene *PgGRXC9* as the candidate stress resilience root growth regulator. Functional characterization of its closest *Arabidopsis* homolog *AtROXY19* revealed a novel role for this glutaredoxin (GRX) gene clade in regulating cell elongation. In summary, our study suggests a conserved function for GRX genes in conferring root cell elongation and enhancing resilience of pearl millet to its Sahelian environment.

## eLife assessment

This is an **important** paper that combines methods ranging from agronomy and plant breeding to *Arabidopsis* functional genetics, to argue that polymorphism in a single gene affects crop yield in pearl millet by affecting root cell elongation and drought stress resilience in a poorly studied crop. The overall argument is plausible but whether the **solid** evidence generated with *Arabidopsis* experiments can be extended to pearl millet itself is unclear.

**eLife digest** Pearl millet is a staple food for over 90 million people living in regions of Africa and India that typically experience high temperatures and little rainfall. It was domesticated about 4,500 years ago in the Sahel region of West Africa and is one of the most heat and drought tolerant cereal crops worldwide.

In most plants, organs known as roots absorb water and essential nutrients from the soil. Young pearl millet plants develop a fast-growing primary root, but it is unclear how this unique feature helps the crop to grow in hot and dry conditions.

Using weather data collected from the Sahel over a 20-year period, Fuente, Grondin et al. predicted by modelling that early drought stress is the major factor limiting pearl millet growth and yield in this region. Field experiments found that plants with primary roots that grow faster within soil were better at tolerating early drought than those with slower growing roots.

Further work using genetic approaches revealed that a gene known as *PgGRXC9* promotes the growth of the primary root. To better understand how this gene works, the team examined a very similar gene in a well-studied model plant known as *Arabidopsis*. This suggested that *PgGRXC9* helps the primary root to grow by stimulating cell elongation within the root.

Since it is well adapted to dry conditions, pearl millet is expected to play an important role in helping agriculture adjust to climate change. The findings of Fuente, Grondin et al. may be used by plant breeders to create more resilient and productive varieties of pearl millet.

## Introduction

Pearl millet was domesticated about 4500 years ago in the Sahelian part of West Africa (*Burgarella et al., 2018*) and is one of the most heat and drought tolerant cereal crops (*Debieu et al., 2017*; *Varshney et al., 2017*). Today, it is the sixth cereal in terms of world production, and it is mostly cultivated in arid regions of sub-Saharan Africa and India where it plays an important role for food security. However, in Africa, pearl millet yield remains low compared to its genetic potential because it is mostly cultivated in marginal lands in low-input and rainfed agricultural systems and the development and adoption of improved varieties is still limited (*Olodo et al., 2020*).

The plant root system is responsible for water and nutrient acquisition from the soil. Breeding for root traits that could improve the crop root system efficiency has been proposed as one of the pillars of a second green revolution (*Den Herder et al., 2010*; *Lynch, 2007*; *Lynch, 2019*). Improved crops with optimized soil resources acquisition might be particularly relevant in low-input and rainfed agrosystems found in the Sahelian region of Africa (*Ndoye et al., 2022*). This strategy relies on the selection of root traits suitable for the specific characteristics of the target environment such as soil and climate but also agricultural practices (*Lynch, 2019*; *Ndoye et al., 2022*; *van der Bom et al., 2020*). It requires a better understanding of stress patterns and the performance of individual root traits in real conditions and in response to different constraints. However, only a few studies have addressed the importance of individual root traits in field conditions.

Primary root development is an important contributor to seedling vigor and greatly influences plant establishment (*Peter et al., 2009*). Pearl millet embryonic root system development is characterized by the formation of a fast-growing primary root that is the only architectural component of the root system for the first 6 days after germination (DAG) (*Passot et al., 2016*). No seminal roots are present in pearl millet and crown and lateral roots were only observed 6 DAG (*Passot et al., 2016*). We hypothesized that the fast-growing primary root might be an adaptation to the Sahelian environment (*Passot et al., 2016*). Here, we show that early drought stress after germination is a major constraint in Sahelian environments and that primary root growth is correlated with increased tolerance to this stress under field conditions in pearl millet. Differences in root growth seem to be mainly regulated by changes in cell elongation. A combination of genome-wide association study (GWAS) and bulk segregant analysis (BSA) on a bi-parental population identified one genomic region controlling this trait. Combining RNAseq, re-sequencing, and re-annotation of this region, we identified a glutaredoxin-encoding gene, *PgGRXC9* as a potential candidate regulator. Functional characterization of the closest homolog in *Arabidopsis* reveals a new role for GRX in the regulation of root growth through cell elongation in the root apical meristem.

## Results

### Early drought stress episodes are an important constraint in Sahelian agrosystems

We previously hypothesized that the fast-growing primary root might be an adaptation to the Sahelian environment and, in particular, to early cycle drought stress episodes (*Passot et al., 2016*). To analyze the frequency and impact of such early drought stress, we first studied meteorological data collected for the past 21 years (2000–2020) at the CNRA Bambey station, located at the center of the pearl millet growing region of Senegal and highly representative of the climate found in Sahelian West Africa. Crops such as pearl millet are traditionally sown before or shortly after the first rain event of the rainy season. Moisture from the first rain event is used by seed to germinate and initiate their growth. We observed frequent intervals between the first and the second significant rainfall event (>10 mm) that could last up to 40 days (*Figure 1—figure supplement 1*). These periods were unrelated to the timing of the first rain event.

A crop model was parameterized using soil and meteorological data (2000–2020) from the Bambey experimental station to determine when pearl millet faced drought stress and predict its impact on crop yield. Using a widely cultivated variety of pearl millet (Souna3) for modeling, we simulated the daily fraction of transpirable soil water (FTSW) profiles in different years to identify periods of the growth cycle when access to water was limiting. FTSW below 0.3 was considered a stressful condition as it is the value below which transpiration rate in pearl millet starts to drop due to insufficient water supply from the root to support transpiration (*Kholová et al., 2010*). Clustering the trend of FTSW in the 21 years studied (based on the methodology of *Charrad et al., 2014*) revealed three stress patterns: early-stress, late-stress, and no-stress (blue, red, and green lines in *Figure 1A*). The crop faced early stress at a frequency of 24% over the 21 years, which resulted in an average 43% grain yield penalty compared to years without stress. Biomass production (an important source of fodder for livestock) was also reduced by 44% on average. Late-stress occurred in 19% of the years and resulted in 25% and 12% of yield and biomass penalty, respectively. No-stress was observed in 57% of the years. Hence, our analysis confirms that early drought episodes are a major constraint in Sahelian agrosystems of West Africa affecting crops during the vegetative stage.

### Primary root growth is correlated with increased tolerance to early drought stress

In order to test if a fast-growing primary root after germination might be an adaptive trait to early drought stress, we first phenotyped a diversity panel of pearl millet inbred lines representative of the genetic diversity of the crop (*Debieu et al., 2018*). As primary root growth is linear in the first days of growth (*Passot et al., 2016*), root length was measured as a proxy of growth on a paper-based hydroponic system 6 DAG in 122 pearl millet inbred lines (5 plants/line in average for a total of 853 plants). Primary root length ranged from 21.1 to 193.2 mm with an average of 94.4 mm and a standard deviation of 32.5 mm, thus showing that a large diversity was available for this trait (*Figure 1B*).

The importance of early root growth for plant tolerance to early drought stress was then evaluated in field trials in 2 independent years using nine inbred lines with contrasted root growth (slow, intermediate, and fast primary root growth). Field trials were set up during the hot dry season (March–June) in 2018 and 2020. Seeds were sown and irrigation equivalent to a 30-mm rainfall was applied. Irrigation was then stopped to mimic an early drought stress episode and plant growth and ecophysiological parameters were analyzed for the following 6 weeks (*Supplementary file 1*, Table S1). We then analyzed the correlations between primary root length, measured in the lab, and plant performance measured in field conditions. Principal component analyses revealed a covariation of early primary root growth and stay-green (the % of leaves that remained green) as well as the performance index of photosynthesis (PI) measured both in 2018 and 2020 (*Figure 1—figure supplement 2*). Correlations between variables adjusted means across the two field trials showed a strong and significant relationship between early primary root growth and stay-green at the vegetative stage, an indicator of plant tolerance to drought stress ($R = 0.95$, p <0.01; *Figure 1C, D*). Hence, our field experiments support the hypothesis that rapid growth of the primary root is beneficial for pearl millet plants to cope with early drought stress episodes after germination.

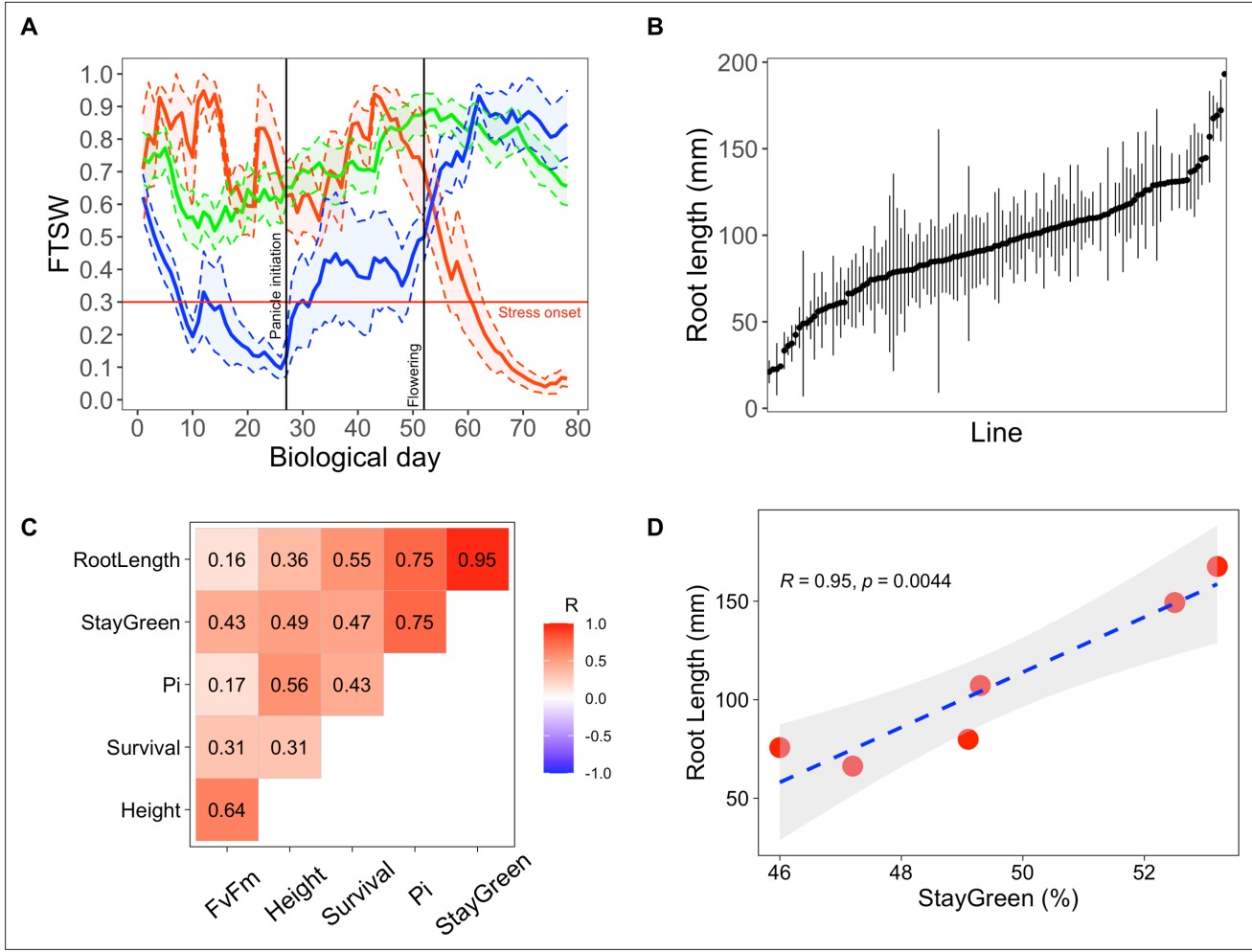

**Figure 1.** Early primary root growth and its correlation to drought tolerance related traits measured in field conditions. (**A**) Stress patterns identified by clustering of simulated fraction of transpirable soil water (FTSW) trend during crop growth in 2000–2021 in Bambey. FTSW = 0.3 (horizontal red line) was considered as the onset of water stress. Each point is the average of a few daily FTSW based on the biological day which was used instead of daily FTSW to highlight the critical stages (panicle initiation and flowering, black lines). Thick lines and shaded areas show the mean and two times standard error (SE), respectively. Blue, red, and green lines correspond to early-stress, late-stress, and no-stress, respectively. (**B**) Root length in 122 pearl millet inbred lines derived from West and Central African landraces. Root length was measured at 6 days after germination in a paper-based hydroponic system installed in a growth chamber. Points represent mean ± SE. (**C**) Correlation between root length and the field measured traits using adjusted lsmeans across both years. The Pearson correlation coefficients are indicated for each pair of traits. (**D**) Linear regression between root length and stay-green. Red points represent the lsmean for the six inbred lines that were common between the two field trials.

The online version of this article includes the following figure supplement(s) for figure 1:

**Figure supplement 1.** Lapse between the first and the second significant rainfall.

**Figure supplement 2.** Covariation of root length, plant height, and drought related traits including maximum quantum efficiency of photosystem II (FvFm) and performance index of photosynthesis (PI) measured at 32 days after sowing, stay-green and plant survival measured at 42 days after sowing.

## Identification of genomic regions in pearl millet controlling early primary root growth

We next studied the genetic determinants of primary root growth in pearl millet. First, the heritability of early primary root growth was analyzed using our paper-based phenotyping data on the panel of inbred lines. Heritability of 0.53 was computed indicating that early primary root growth is under strong genetic control in pearl millet. We therefore conducted a GWAS. Genotyping by sequencing of the panel of pearl millet inbred lines provided 392,493 single nucleotide polymorphisms (SNPs) for association, after filtering on quality, with an average density of 2.5 SNPs per 10 kb (*Debieu et al., 2018*). For the current study, a set of 392,216 SNPs polymorphic for the 122 inbred lines with a

phenotype was selected to conduct GWAS (*de la Fuente Cantó et al., 2022*). GWAS was performed using the ridge latent factor mixed model (LFMM) algorithm (*Caye et al., 2019*). In addition, we considered other GWAS methods to contrast the results (analysis of variance [ANOVA], efficient mixed model association [EMMA], or mixed linear model [MLM]; *Figure 2—figure supplement 1*). Our analysis revealed a total of 447 significant SNPs across the pearl millet genome associated with primary root growth from which, 109 SNPs were found highly significant with at least two other methods for association analysis at p-value <$10^{-4}$ (*Supplementary file 1*, Table S2). Only two of these markers, located on chromosomes 1 and 3, were above the 0.05 false discovery rate significance threshold (*Figure 2A*).

To validate our GWAS analysis, we generated a bi-parental population from two inbred lines with contrasting early primary root growth. The two lines, ICML-IS 11155 (low primary root growth) and SL2 (high primary root growth), were selected based on our initial paper-based root phenotyping experiment. The difference in primary root growth between these two lines was confirmed 7 DAG in a paper roll phenotyping system (p <0.001; *Figure 2—figure supplement 2*), as well as in soil using X-ray microCT 10 DAG (p = 0.058 but with a low number of samples analyzed; *Figure 2—figure supplement 2*), thus demonstrating their contrasting primary root phenotypes were robust and independent of the experimental system.

Lines SL2 and ICML-IS 11155 were crossed and 737 F2 plants were phenotyped for early root growth together with their parents (33 ICML-IS 11155 and 30 SL2 plants). The phenotypes of F2 plants showed a normal distribution encompassing the range of phenotypes from the two parents (*Figure 2—figure supplement 3*). 75 F2 plants were selected for each extreme phenotype (75 highest and 75 lowest growth) and were used for bulk DNA extraction. The corresponding DNA was then used for genotyping by sequencing. Mean average sequencing depth in the bulks corresponded to 1028X (high growth) and 814X (low growth). After filtering, a group of 33,582 SNP variants (2.1 SNP per 100 kb in average) identified between the bulks was used to assess the differences in allele frequency linked to the root length phenotype. BSA revealed differences in allele frequency for 1285 SNPs (*Figure 2B*; *Supplementary file 1*, Table S3). Six regions consisting of clusters of neighboring markers with overlapping region of significance defined by simulations (*de la Fuente Cantó and Vigouroux, 2022*) and equivalent to ±8 Mbp around each significant marker were identified on chromosome 1 (RL1.1, RL1.2, and RL1.3) and chromosome 6 (RL6.1, RL6.2, and RL6.3; *Figure 2B*). Eighteen marker–trait associations identified by GWAS co-localized with these BSA regions of significance (*Supplementary file 1*, Table S4) including the most significant GWAS SNP on chromosome 1 at position 231264526 (*Figure 2A*).

Root growth is dependent on cell division and cell elongation activities occurring at the root tip. To identify the cellular process responsible for changes in root growth between the two parental lines, ICML-IS 11155 and SL2, we used confocal microscopy to image and measure cell elongation in the root tip starting from the quiescent center (*Figure 2C*, *Figure 2—figure supplement 4*). While cell production rates were similar in both lines (*Figure 2D*), SL2 (fast growth line) exhibited a significantly higher cell elongation rate than ICML-IS 11155 (*Figure 2E*). Hence, the significant difference in root growth between inbred lines SL2 (fast growth) and ICML-IS 11155 (slow growth) was mainly driven by changes in root cell elongation.

As differences in root growth between the two parent lines were mainly driven by cell elongation, we hypothesized that it might be linked to genes that are expressed in the root tip. We therefore profiled gene expression using RNAseq in the primary root tip (2 cm apex) of inbred lines ICML-IS 11155 (slow growth) and SL2 (fast growth). Reads were mapped to the coding DNA sequences (CDSs) predicted in the reference genome (*Varshney et al., 2017*) for expression analyses. 1778 genes showed significant differences in gene expression between the two contrasted lines using three combined statistical tests (EdgeR, DESeq, and DESeq2, p-value <0.01). Unexpectedly, a large proportion of the reads (31.48% in average in the six RNAseq experiments) did not map to predicted genes on the reference genome. When further analyzed, 15.15% of the unmapped reads (with no correspondence to predicted CDS) were found not to match the reference genome. These might correspond either to unsequenced regions or to genotype-specific genomic regions that are not present in the reference line. The remaining unmapped reads corresponded to either rRNA and tRNA genes (40.28% of the unmapped reads) or to non-annotated genes or non-coding RNAs (44.57% of the unmapped reads). Hence, our transcriptomics analysis identified genes that are differentially expressed in the root tip

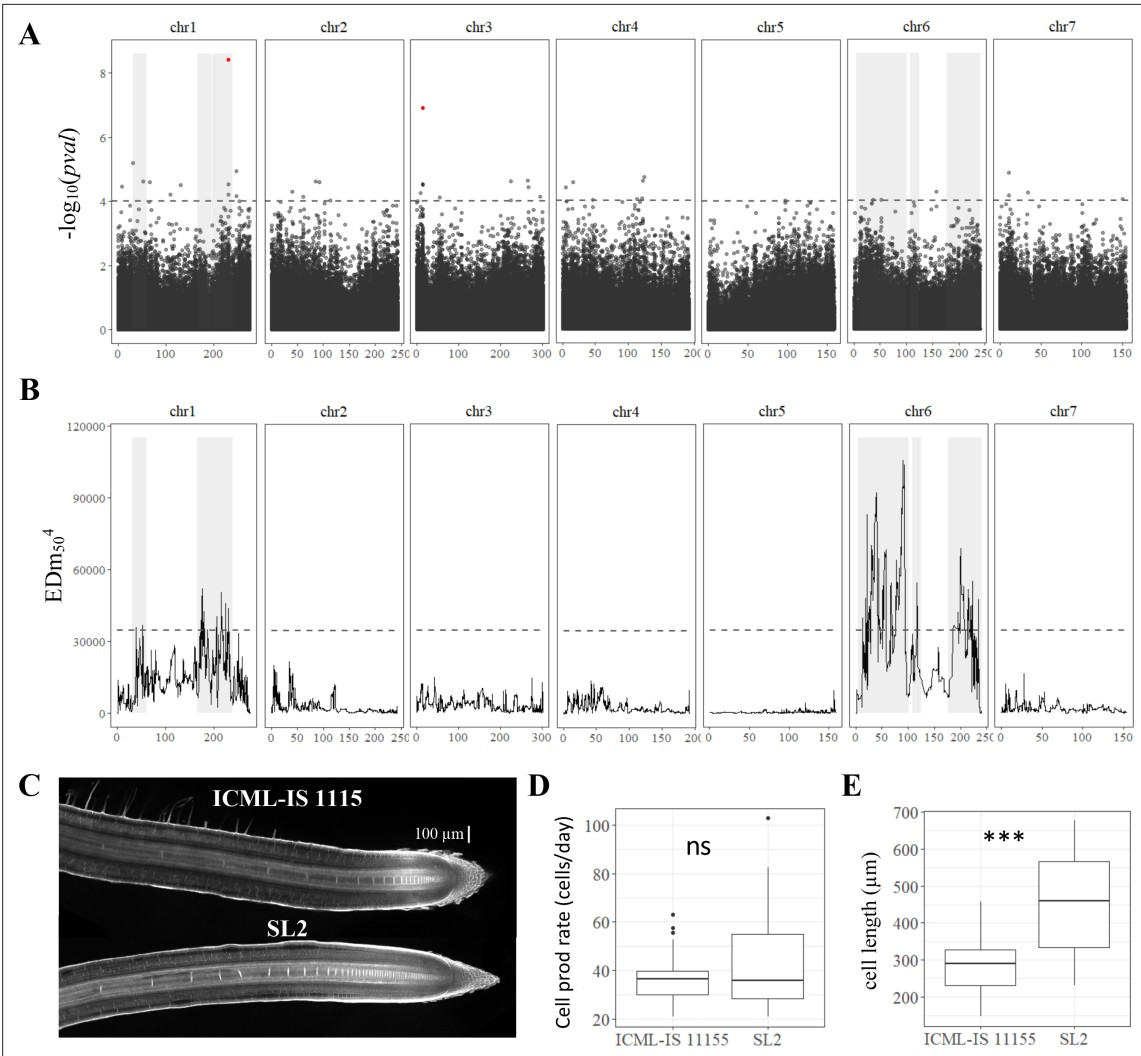

**Figure 2.** Genetic dissection of primary root length in pearl millet and analysis of root apical meristem and primary root growth of contrasted pearl millet lines. (**A**) Manhattan plot of the genome-wide association study (GWAS) by lfmm ridge method (**Caye et al., 2019**). The horizontal axes correspond to the map position of each of the 392,216 SNPs identified by GBS in a group of 122 inbred lines. The vertical axis indicates the $-\log_{10}$ p-value of the statistic. The dash line delimits the threshold for highly significant SNPs (p-value $<10^{-4}$). Significant SNP markers above the 0.05 false discovery rate (FDR) significance threshold are highlighted in red. (**B**) Bulk segregant analysis (BSA) identification of significant regions associated with primary root length using bulks of contrasted F2 lines from a bi-parental cross. The plot shows the Euclidean distance statistic profile (y-axis) across the seven pearl millet chromosomes (x-axis). The dash line indicates the 95% confidence interval threshold for the localization of significant regions. In both plots, the shaded area delimits the extent of the six significant regions identified by BSA and therefore the overlap with significant SNPs identified by GWAS (**A**) and the correspondence with the BSA peaks found (**B**). (**C**) Confocal image of the root tip of the contrasted inbred lines for primary root growth (ICML-IS-1115 and SL2 with slow and fast growth, respectively). (**D**) Estimation of cell production rate for ICML-IS-1115 (N = 25) and SL2 (N = 22) according to **Beemster et al., 2002**. (**E**) Maximum cell length reached in the root elongation zone for ICML-IS-1115 (N = 25) and SL2 (N = 22). ***p-value ≤0.0001, ns: not significant.

The online version of this article includes the following figure supplement(s) for figure 2:

**Figure supplement 1.** Genome-wide association study (GWAS) Manhattan plot and p-values QQ plots obtained with five methods.

**Figure supplement 2.** Contrasted primary root length of lines ICML-IS 11155 and SL2.

**Figure supplement 3.** Distribution of root growth phenotypes in the F2 population and the parents ICML-IS 11155 and SL2.

**Figure supplement 4.** Cell length profile in the root meristem of (**A**) SL2 and (**B**) ICML-IS 11155.

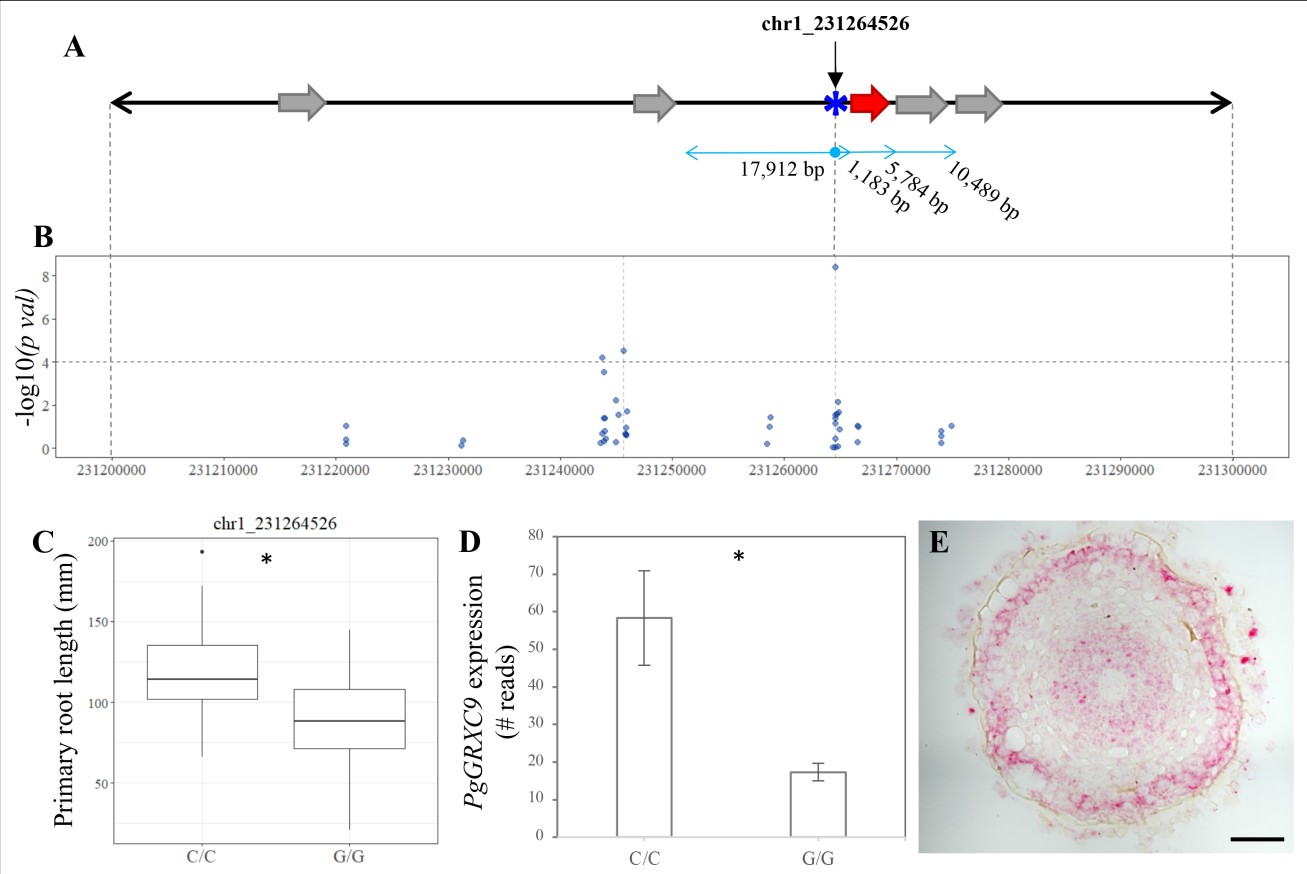

**Figure 3.** Identification of candidate genes for primary root length in pearl millet. (**A**) Annotated genes in 100 kb region of chromosome 1 harboring significant SNPs identified by genome-wide association study (GWAS) and coincident with a bulk segregant analysis (BSA) significant region. Blue asterisk shows the position of the SNP with the most significant association. Gray arrows represent the predicted genes according to the reference genome (*Varshney et al., 2017*). The red arrow represents the new predicted gene, PgGRXC9, identified by re-annotation of the genomic interval using *de novo* pearl millet assembly based on contig sequences. (**B**) GWAS Manhattan plot in the 100 kb interval on chromosome 1. Dots represent the $-\log_{10}$ (pval) of the lfmm ridge statistic (*y*-axis) for the SNPs identified in the region. The dashed horizontal line shows the threshold for highly significant SNPs. Positions and physical distance between genes and markers are displayed in bp. (**C**) Primary root length phenotype associated to the allelic variants of the two significant SNPs in the region: chr1_231264526 (NC/C = 26, NG/G = 74). (**D**) Expression of *PgGRXC9* for the two alleles from the RNAseq data. *p-value ≤0.01. (**E**) Transversal section in the elongation zone of a primary root showing the expression profile of *PgGRXC9* as indicated by RNAscope. The localization of this section and the expression of *PgGRXC9* in other part of the root is indicated in *Figure 3—figure supplement 1*. Scale bar = 50 mm.

The online version of this article includes the following figure supplement(s) for figure 3:

**Figure supplement 1.** Expression pattern of *PgGRXC9* in the root tip as revealed by RNAscope.

of inbred lines ICML-IS 11155 and SL2 but revealed that some genes expressed in root tips are not annotated in the current version of the pearl millet genome.

## Re-sequencing of the root length QTL region reveals a new GRX-encoding gene

We re-analyzed a 1-Mbp genomic region on chromosome 1 around the most significant GWAS marker–trait association, corresponding to SNP chr1_231264526, that co-localizes with a BSA QTL. This region contained a large proportion of unknown nucleotides (16.39% N) thus making gene annotation difficult. To obtain better quality sequence information, long reads (Nanopore technology, *Yuan et al., 2017*) corresponding to the target QTL regions were recovered and re-annotated using structural and gene expression (RNAseq) data. Re-annotation revealed one novel 465 bp CDS, 1103 bp downstream of the significantly associated SNP (chr1_231264526, *Figure 3A, B*). This new CDS encodes a protein

with strong homology to glutaredoxin (GRX) C9-like proteins from various cereals and was named *PgGRXC9*.

Two alleles at this locus exist in our population (C or G) with plants carrying a homozygous C/C allele (21.3% of the lines in our population) having a significantly higher root growth than plants carrying the G/G homozygous allele (60.7% of the lines; *Figure 3C*). We found that *PgGRXC9* expression was significantly higher in the root tip of the line carrying the allele associated with higher root growth (C/C) compared to the line carrying the lower growth allele (G/G; *Figure 3D*). *In situ* hybridization using the RNAscope technology revealed expression in the stele and epidermis in the root tip and elongation zone (*Figure 3E*, *Figure 3—figure supplement 1*). Expression was also observed in the columella cells of the root cap and in the stele of the differentiated part of the root (*Figure 3—figure supplement 1*). We then used the RNAseq data to search for polymorphisms between the two contrasted parent lines used for BSA. No sequence polymorphisms were found in the predicted coding sequence of the gene between the two parent lines. However, two polymorphisms were detected in the 5′UTR region. Altogether, our results suggest that *PgGRXC9* is a positive regulator of root growth and that a polymorphism in the promoter region of *PgGRXC9* might led to changes in its expression level and ultimately to a quantitative difference in root growth between the two lines.

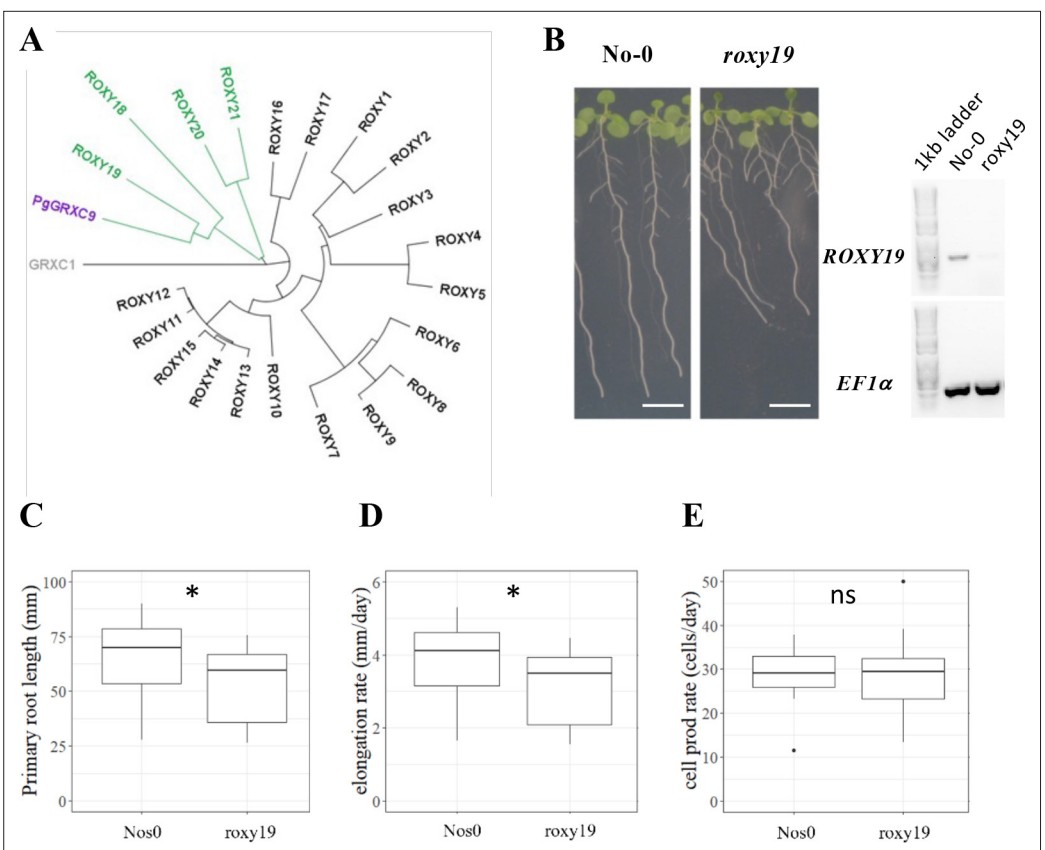

**Figure 4.** The *Arabidopsis* homolog of PgGRXC9 regulates root growth. (**A**) Phylogenetic tree of ROXYs protein sequences. This tree has been obtained from pairwise alignments of all whole protein sequence pairs using the Neighbor-Joining method and Jukes–Cantor distance matrix. GRXC1 (gray), a member of Class I GRX, is used as the tree root. *Arabidopsis* ROXY18-21 are represented in green, and PgGRXC9 is represented in purple. (**B**) Contrasted primary root phenotype in Nos0 and roxy19 in 12-day-old seedlings grown in agar plates. Scale bar: 1 cm. (**C**) Primary root length, (**D**) cell elongation rate, and (**E**) estimation of the cell production rate assuming a steady root growth rate of Nos0 (*N* = 27) and roxy19 (*N* = 25) plants. *p-value ≤0.01, ns: not significant.

The online version of this article includes the following source data and figure supplement(s) for figure 4:

**Source data 1.** *ROXY19* and *EF1a* (control gene) expression detected by reverse transcription PCR in different *Arabidopsis* lines.

**Figure supplement 1.** Protein alignment of PgGRXC9 and *Arabidopsis* ROXY members.

However, the effect of the polymorphisms in the promoter on gene expression needs to be tested to validate this hypothesis.

## *PgGRXC9 Arabidopsis* ortholog *ROXY19* also regulates root cell elongation

To test the hypothesis that changes in *PgGRXC9* expression level are responsible for a change in root growth, we studied its ortholog in the model plant *Arabidopsis thaliana* (as functional genomic studies are not possible in pearl millet). The closest homolog in *Arabidopsis* was *ROXY19* (*AT1G28480*) with 56% identity at the protein level (*Figure 4A*; *Figure 4—figure supplement 1*). The ROXY family is a land plant-specific family of GRX, with 21 members in *Arabidopsis* (*Meyer et al., 2009*). They all harbor a Cys-Cys (CC) putative active site and bind to TGA type transcription factors via the C-terminal domain (*Meyer et al., 2012*). Several of them contain a final ALWL C-terminal motif responsible for binding to TOPLESS and TOPLESS-related transcriptional co-repressors (*Uhrig et al., 2017*). All these important amino acids were conserved in PgGRXC9 (*Figure 4—figure supplement 1*). ROXY19 belongs to a subgroup of four ROXYs (ROXY18-21) that contain a specific N-terminal domain of unknown function, that was also present in PgGRXC9 (*Figure 4—figure supplement 1*).

Transcriptomic data indicate that *ROXY19* is expressed in root tissues, with the strongest expression detected in the columella and stele of the elongation and maturation zone and some expression in the ground tissue and epidermis (*Belin et al., 2015*). Given this expression profile was similar to *PgGRXC9* in pearl millet, we exploited a *roxy19* null mutant available in the *Nossen* (No-0) background (*Huang et al., 2016*) to study its function. Strikingly, we observed that *roxy19* had reduced primary root growth compared to its wild-type ecotype (No-0; *Figure 4B, C*). Closer examination revealed that the defect in *roxy19* root growth was due to a reduction in cell elongation (*Figure 4D*), whereas cell production rate was similar in the mutant and wild type (*Figure 4E*), thus mimicking the phenotype observed in pearl millet. Hence, our functional studies reveal that *ROXY19*, the closest homolog of *PgGRXC9* in *A. thaliana*, is a positive regulator of root growth through modulation of cell elongation.

## Discussion

Drylands, defined as regions where precipitation is lower than atmospheric water demand, cover around 40% of the land surface and host about 2 billion people (*Wang et al., 2022*). They are key regions for global food security as it is estimated that they are responsible for about 60% of global food production (*Wang et al., 2022*). They also play an important role for the global carbon budget (*Wang et al., 2022*). Climate change is expected to have a negative impact on agriculture in drylands with direct consequences for the livelihood of about 178 million people by 2050 under a 1.5°C temperature increase scenario (*Wang et al., 2022*). There is therefore an urgent need to devise new agricultural practices and crop varieties to address this challenge. This is particularly critical in the Sahel, the dryland region stretching across Africa and delimited by the Sahara to the North and the Sudanian savanna to the South, where agriculture is mainly rainfed with limited access to fertilizers and irrigation (*Ndoye et al., 2022*).

Pearl millet was domesticated in the Sahel about 4500 years ago (*Burgarella et al., 2018*) and it is a key crop for food security in that region (*Debieu et al., 2017*; *Varshney et al., 2017*). In the Sahel, drought stress is a major factor limiting crop yield. The semi-arid tropical climate is characterized by a long dry season and a short rainy season where most of the agriculture occurs. Pearl millet is traditionally sowed before or right after the first rain so that the water from this first rain is used for germination and seedling establishment. However, the precipitation pattern is irregular and varies from season to season and these intra- and interannual variability are expected to increase with future climate (*Sultan and Gaetani, 2016*). To devise strategies to adapt agriculture to future climate, it is important to identify the stress patterns faced by crops. In this study, we used meteorological data together with a crop model to estimate the water stress patterns faced by pearl millet in the 2000–2020 period. This revealed three main types of patterns, early drought stress, terminal drought stress, and no stress. Early drought stress, corresponding to rain pause during crop establishment, was found to occur roughly one fourth of the years and to have a large impact on both grain and biomass (an important source of fodder) production. Indeed, we found that lapse between the first rain and subsequent significant rain were occurring frequently, and current models predict that such gaps are going to be

more frequent in future climate (*Sultan and Gaetani, 2016*). This indicates that early drought stress after germination is a major constraint for crop growth in the Sahelian agrosystems.

The embryonic root system makes most of the root system for the first weeks of the seedlings life and is therefore important for crop establishment. Interestingly, we previously reported that early root system development in pearl millet is characterized by a unique fast-growing primary root (*Passot et al., 2016*). There are no seminal roots, and post-embryonic root system branching (formation of lateral and crown roots) only starts 6 DAG (*Passot et al., 2016*). We hypothesized that this could be beneficial for early establishment and access to deep resources (water and nutrients) as an adaptation to the Sahelian climate and, in particular, to face early drought stress events. We used the natural variability for primary root growth in pearl millet to test this hypothesis in 2 years of field trials set up to mimic early rain pause episodes after germination. We observed that pearl millet lines with faster primary root growth had better tolerance to this stress thus indicating that indeed fast primary root growth was beneficial to cope with post-germination drought stress.

Early primary root growth showed a high heritability. We therefore studied the genetic bases of this useful trait using a combination of association genetics (GWAS) and QTL analysis (BSA). We are confident our genetic analysis targeted specifically early primary root growth rather than seed reserves or seedling vigor because (1) we previously showed that primary root growth was not correlated to seed weight in our experimental setup (*Passot et al., 2016*) and (2) we selected parents with very contrasted root growth but similar shoot biomass to generate the bi-parental population used for QTL analysis. Moreover, the lines used for QTL analyses showed significant differences in primary root growth in different root phenotyping systems (including soil columns) thus suggesting that the loci we identified are robust and relevant for primary root growth *in naturae*. Our genetic analysis revealed a limited number of loci controlling early primary root growth.

We focused our analysis on the most significant marker–trait association for GWAS that was co-localizing with a BSA QTL on chromosome 1. Re-sequencing and re-annotation of the corresponding genomic region revealed a new gene close to the most significant SNP. This gene, *PgGRXC9*, encodes a potential glutaredoxin protein. The gene is more expressed in the root tip of the line with the allele associated with higher root growth. We found no polymorphism in the *PgGRXC9* coding region in the two parents of the QTL population. We showed that the closest homolog in *A. thaliana*, *AtROXY19*, has a similar expression pattern and that it regulates primary root growth. Interestingly, the root growth phenotypes observed in contrasted pearl millet lines and in the *roxy19* mutant indicate that, in both cases, regulation of root growth occurs at the cell elongation level. Although the link between PgGRXC9/AtROXY19 and root growth elongation is novel, previous studies have revealed connections between root redox status and growth. In *Arabidopsis*, other members of the ROXY family have been shown to regulate root growth elongation in response to nitrate (*Ota et al., 2020*; *Patterson et al., 2016*). More generally, redox homeostasis has been previously reported to regulate root meristem organization and functioning in *Arabidopsis* (e.g., *Bashandy et al., 2010*; *Tsukagoshi, 2016*; *Tsukagoshi et al., 2010*; *Vernoux et al., 2000*) including regulation of cell elongation (*Mabuchi et al., 2018*; *Tsukagoshi, 2016*). However, the regulatory role of ROXY19 on root cell elongation (currently based on the analysis of a single mutant allele) needs to be confirmed and the mechanism and the actors of the regulation of root elongation by ROXY19 will need further investigation.

Based on our results, we propose that redox regulation in the root meristem might be responsible for a root growth QTL in pearl millet. Indeed, our data suggest that changes in *PgGRXC9* expression level due to polymorphisms in the promoter or 5'UTR region of the gene level might cause differences in root cell elongation and ultimately root growth that are important for adaptation to post-embryonic drought stress. However, the function of *PgGRXC9* in cell elongation, its site of action and its molecular targets still need to be demonstrated. Furthermore, the implication of the redox is only suggested by indirect evidence and needs to be explored more directly. Further work will therefore be needed to validate this hypothesis. Pearl millet is an orphan crop and no efficient functional genomics tools are currently available. Future work will first target the development of an efficient gene editing protocol in this species.

## Ideas and speculation

Pearl millet evolved and was domesticated in the Sahel. Its seeds are small (8 mg on average, about 5 and 20 times lighter than wheat or maize seeds, respectively, for example) with limited reserves

available for early seedling growth. It invests the available seed reserves toward the growth of a unique primary root to rapidly colonize deeper soil layers rather than forming more root axes (seminal roots for example) as seen in other cereals. Based on our results, we speculate that this specific early root development strategy was selected during evolution to cope with the specific rain pattern encountered in the Sahel.

## Materials and methods

### Plant materials

The panel of pearl millet inbred lines derived from West and Central African landraces (open-pollinated varieties) used in this study has been previously described (*Debieu et al., 2018*). Nine lines from this panel that were contrasting for root length were selected for field trials (*Supplementary file 1*, Table S5).

### Field trials

Field experiments were performed at the CNRA station (Centre National de Recherche Agronomique) of the Institut Sénégalais des Recherches Agricoles (ISRA) in Bambey, Senegal (14.42°N, 16.28°W), during the dry season of 2018 and 2020 to fully control irrigation. Fields are composed of deep sandy soil with low levels of clay and silt (12%) and organic matter (0.4%). Clay and silt content increase with soil depth from 10.2% in the 0–0.2 m layer to 13.3% in the 0.8–1.2 m layer. Experiments were set up using a complete randomized block design with 4 plots per variety, each composed of 6 rows of 6.3 m long with 0.9 m between plants and 0.9 m between rows (42 plants/plot). Irrigation was provided after sowing (30 mm of water) to allow seeds to germinate and was followed by a period of 42 days of water withholding to impose seedling drought stress. Thinning was performed 15 days after sowing to conserve a single plant per planting hole. Fertilization (NPK) following standard recommendation of 150 kg ha$^{-1}$ NPK (15-15-15) was applied to the entire trial after sowing and before irrigation. Fields were maintained free of weeds. Plant height was measured at 42 days after sowing. Stay-green trait expressed as the percentage of green leaves compared to the total number of leaves was estimated on 3 plants per plot at 42 days after sowing. Survival rate was measured as the percentage of surviving plants at 42 days after sowing in each plot compared to the initial number of plants that had emerged. Photosynthesis parameters (FvFm: maximum quantum efficiency of photosystem II and PI: performance index of photosynthesis) were measured on three plants per plot at 32 days after sowing in both 2018 and 2020 using a Handy Pea chlorophyll fluorometer (Hansatech Instruments Ltd).

### Identification of water stress pattern

Long-term weather data (2000–2021) in CNRA (Bambey, Senegal) were gathered and analyzed to evaluate how the rainfall gap affects crop growth. An adapted version of the Simple Simulation Model (SSM-iCrop, *Soltani et al., 2013*; *Soltani and Sinclair, 2012*) was used to simulate the crop growth of a common West African pearl millet genotype (Souna3) and dynamic of water in the soil. To test the effect of water limitation alone, the model was run in water-limited potential mode with 3.7 plants per m$^{-2}$ density and no fertilizer limitation. From April 10, the model began simulating soil water balance with a quarter-saturated profile. Sowing date was defined by the first significant rain of the year (actual transpirable soil water superior or equal to 10 mm). Clustering was done based on the trend of the FTSW from sowing date to maturity in different years to group years into different stress patterns and the effect on the yield and biomass. The NbClust Package (*Charrad et al., 2014*) was used to determine the optimal number of clusters and the dynamic time warping method to cluster the daily simulated FTSW in different years.

### Root growth phenotyping

For high-throughput experiments (association genetics and BSA), plants were phenotyped for primary root growth with a paper-based hydroponic system as previously described (*Passot et al., 2016*). Seeds were surface sterilized and pre-germinated in Petri dishes, transferred into pouches 24 hr after germination at a density of 3 seeds per paper and then maintained in a growth room with a 14-hr photoperiod (28°C during day and 24°C during night). Pictures of the root systems were taken 6 DAG with a D5100 DSLR camera (Nikon) at a resolution of 16 M pixels. The camera was fixed on a holder

to maintain the same distance between the lens and each root system. Primary root lengths were measured using RootNav (**Pound et al., 2013**). Rhizotron experiments were performed as previously described (**Passot et al., 2018**). For X-ray tomography, seeds were sterilized with 20% bleach for 5 min, then washed with sterilized water five times. Sterilized seeds were pre-germinated on moist sterilized filter paper for 36 hr at 28°C in a plant growth chamber (12-hr photoperiod and 300 μmol/m²/s light with 70% relative humidity). Equally germinated pearl millet seedlings (1 cm radicle length) were gently placed in loamy sand soil in CT columns (30 cm height × 5 cm diameter). Loamy sand soil collected from the University of Nottingham farm at Bunny, Nottinghamshire, UK (52·52°N, 1·07°W) was crushed thoroughly and sieved through 2-mm mesh size. These columns were saturated with water and then drained to field capacity. Five replicates of each SL2 and ICML-IS 11155 pearl millet seedlings were grown for 10 days for the CT experiment in a growth chamber maintained at a 12-hr photoperiod at 25°C and 250 μmol/m²/s light with 60% relative humidity. The root systems of 10-day-old pearl millet lines (ICML-IS 11155 and SL2) were imaged non-destructively using a GE Phoenix v|tome|x M 240 kV X-ray tomography system (GE Inspection Technologies, Wunstorf, Germany). Scans were acquired by collecting 3433 projection images at 140 kV X-ray energy, 200 μA current and 131ms detector exposure time at scan resolution of 45 μm in FAST mode (8-min total scan time). Three-dimensional image reconstruction was performed using Datos|REC software (GE Inspection Technologies, Wunstorf, Germany) and root length was measured using the polyline tool in VGStudioMax (Volume Graphics GmbH, Germany).

For *Arabidopsis* experiments, seeds were surface sterilized and placed on plates containing half-strength Murashige and Skoog (1/2 MS) medium with 0.5 g l⁻¹ 2-morpholineethanesulfonic acid (MES) and 0.8% (wt/vol) plant agar without sucrose. All plates were incubated vertically at 20°C with 160 μE m⁻² s⁻¹ light intensity and a 16-hr light/8-hr dark regime. The primary root elongation rate was quantified between days 8 and 17. Lengths were quantified from pictures using the public domain image analysis program ImageJ 1.52i (https://imagej.nih.gov/ij/) and its NeuronJ plugin.

## Heritability

Broad sense heritability was computed with the following formula:

$$H^2 = \frac{\text{Var(line)}}{\text{Var(line)} + \dfrac{\text{Var(res)}}{n_{\text{plant/line}}}},$$

where

$n_{\text{plant/line}}$ is the average number of plants measured per line,
Var(line) is the variance associated with lines,
Var(res) is the residual variance.

Both variances are parameters of the following linear mixed model:

$$Length = \mu + \alpha_{line} + \varepsilon_{res},$$

where $\mu$ is the overall mean length, $\alpha_{\text{line}}$ is the random effect attached to the lines with $\alpha_{\text{line}} \sim N(0, \text{Var(line)})$, and $\varepsilon_{\text{res}}$ is the error term with $\varepsilon_{\text{res}} \sim N(0, \text{Var(res)})$.

## Genome-wide association mapping

Genotyping by sequencing of the panel of inbred lines was reported in previous work (**Debieu et al., 2018**). In order to conduct association mapping, we first estimated population structure based on the ancestry coefficients estimated with the R package LEA v2.0. Then, GWAS was performed using LFMM 2.0 which corrects for unobserved population confounders and considers ridge estimates (**Caye et al., 2019**). Given the reduced number of lines, GWAS was performed using four other methods for association analysis: ANOVA, EMMA (**Kang et al., 2008**), MLM (**Yu et al., 2006**), and a previous version of LFMM (**Frichot et al., 2013**). The median heterozygosity for the inbred lines was low at 5.6%. Each heterozygous site was randomly fixed for the reference or the alternate allele.

## Bulk segregant analysis

Root growth in F2 seedlings derived from the cross between ICML-IS 11155 and SL-2 (low and high primary root growth, respectively) was characterized in the paper phenotyping system as described above. The 10% extreme lines in the tails of the distribution were selected to form the bulks of contrasted lines. Leaf discs (1.5 mm diameter) were sampled for each line during the phenotyping experiment and stored at −80°C. Leaf discs from selected lines were pooled together to make the high root growth (*H*) and low root growth (*L*) bulks for DNA extraction. Genomic DNA was isolated for each bulk using the MATAB method as previously described (*Mariac et al., 2006*). DNA libraries were constructed from genomic DNA fragmented by acoustic shearing (Bioruptor) with a peak fragment target size of 200–300 bp. Sheared DNA was end-repaired using a T4 polymerase (End Repair NEB) and bound with the P5 and PE-P7 sequencing adaptors (*Rohland and Reich, 2012*). A combination of unique oligonucleotides barcode sequences was ligated to the P5 adaptor to index the DNA libraries derived from each bulk (H and L) and from the parental lines of the cross. Then, equimolar amounts of each DNA library were combined in the genomic DNA bank for the cross. Subsequently, the DNA bank library was hybridized with biotinylated RNA probes or 'baits' (myBaits) targeting the first 500 bp of the 32,100 pearl millet predicted genes. Finally, high-throughput sequencing of the enriched DNA library was performed on an Illumina HiSeq sequencer by Novogene Company Limited (HK). Initial sequencing quality checks using FastQC version 0.11.5 (*Andrews, 2010*) were followed by trimming and quality filter steps on which adaptors, barcode sequences, and low-quality reads (<35 bp) were removed. Filtered reads were aligned to the pearl millet reference genome (*Varshney et al., 2017*) using the Burrows-Wheeler Alignmen tool (BWA version 0.7.17 r1188; *Li and Durbin, 2009*). Reads mapping at the target enriched regions were used for SNP calling using the UnifiedGenotyper algorithm from GATK 3.7 (*McKenna et al., 2010*) with the parameter down-sampling limit (dcov) set at 9000. Multi-allelic sites and those with low total allele frequency (AF <0.25) were removed. In addition, sites with either low or high total sequencing depth (below the 25th and above the 95th percentiles, respectively) were removed. Finally, SNPs with more than 50% missing data and minor allele frequency under 5% were excluded.

For bulk segregant study, only biallelic SNP variants of the bulks that were present in the parental lines of the cross were kept. The line ICML-IS 11155 (low primary root growth) was used as the reference genome for the cross to designate the alternate and reference SNP variants in the bulks. Out of the range of statistical approaches for measuring the differences in allele frequency in BSA, Euclidean distance-based statistics as suggested by *Hill et al., 2013* was selected based on a preliminary study on which we tested the efficiency of the methods for QTL detection using simulations (*de la Fuente Cantó and Vigouroux, 2022*). The Euclidean distance between allele frequencies of the bulks at each marker position (*EDm*) was calculated as follows:

$$EDm = \sqrt{\left(faL - fAL\right)^2 + \left(faH - fAH\right)^2}$$

where *fa* and *fA* correspond to the allele frequency of the alternate and reference allele in the low bulk (*L*) and the high bulk (*H*), respectively. Then, to reduce the effect of sequencing noise and increase the signal of the differences in allele frequency we calculated the fourth power of the cumulative *EDm* value in windows of 100 consecutive markers (*Omboki et al., 2018*; *Zhang et al., 2019*). The confidence interval of the statistic was determined using simulations as described in *de la Fuente Cantó and Vigouroux, 2022*.

## RNAseq

Seeds were surface sterilized and germinated in Petri dishes containing wet filter paper for 24 hr in the dark at 27°C. After 2 days, plants were transferred to hydroponic tanks containing 1/4 Hoagland solution and grown for 15 days at 27°C. RNA was extracted from the root tip (2 cm apex) of the primary root using the RNeasy Plant Mini Kit (QIAGEN). RNAseq was performed by the Montpellier GenomiX Platform (MGX, https://www.mgx.cnrs.fr/). Sequencing was performed on an Illumina HiSeq 2500. Analyses were performed as previously described (*de la Fuente Cantó and Vigouroux, 2022*). Three different statistical tests were used to identify differentially expressed genes: EdgeR (*McCarthy et al., 2012*; *Robinson et al., 2010*), DESeq (*Anders and Huber, 2010*), and DESeq2 (*Love et al., 2014*). GO terms enrichment was performed in the 1778 genes that were significantly differentially

expressed between the slow and fast growth lines according to the three statistical tests using the TopGO package in R. Briefly, overrepresentation of GO terms in the list of differentially expressed genes (1032 of the 1778 with GO annotations) was investigated in relation to the list of annotated pearl millet genes with GO annotations (16,620 genes with GO annotations and 47,234 GO terms in total; *Varshney et al., 2017*) using a Fisher test. Each GO category was tested independently or considering hierarchical links between GO terms. GO terms showing p-values below 0.01 were further considered for enrichment analysis.

## Cellular analysis of root meristems

Root meristem phenotype of lines with contrasted primary root length was characterized using confocal microscopy. Lines ICML-IS 11155 and SL-2 (low and high primary root growth, respectively) were grown in paper rolls under controlled conditions in a growth room. In brief, pre-germinated seeds were sown along the long side of germination paper (Anchor Paper Company, USA) rolled on itself and imbibed in ¼ strength Hoagland solution (*Hoagland and Arnon, 1950*). Each paper roll was placed in an Erlenmeyer flask containing 200 ml of nutrient solution, a volume maintained constant throughout the experiment. Primary root length was measured, and 2 cm length were sampled from the tip of 1-week-old seedlings. Root tips were fixed in FAA solution for 24 hr. Then, root tips were washed twice in PBS solution and moved to the ClearSee clearing (*Kurihara et al., 2015*) solution for a minimum of 48 hr. Before imaging, root tips were stained for 30 min in 0.1% Calcofluor White and washed in ClearSee for other 30 min (protocol adapted from *Ursache et al., 2018*). Root tip images were obtained using a Leica SP8 confocal microscope equipped with a ×20/0.70 dry objective at a detection range of 420–485 nm. Images were analyzed using the Broadly Applicable Routines for ImageJ (*Ferreira et al., 2015*). Cell walls from the meta-xylem vessel were used to estimate cell length along the axial axis of the root. The 'find peaks' option was used to determine the coordinates of the cell walls along a segmented line traced from the quiescent center to the mature zone where the first root hairs were observed, and maximum cell length was reached. Consecutive data points defined the cell length along the root axis and a logistic function was fitted to the data to characterize axial root growth (*Morris and Silk, 1992*). Elongation rate was estimated as the root length reached per day of experiment. This value divided by the maximum cell elongation defined per sample was used to approximate the cell production value or number of cells produced in the meristem per day assuming steady-state growth (*Beemster et al., 2002*).

## Re-annotation of QTL region using long reads data

Long reads were used to reannotate the QTL region as in *Grondin et al., 2020* (Genbank accessions MT474864 and MT474865). For identification of Nanopore contigs corresponding to the QTL region, a 1-MB sequence located around the most significant GWAS SNP was extracted from the reference genome and aligned to the long-read genome using the nucmer tool (MUMmer version 4.0.0beta2, - -mum option *Marçais et al., 2018*) with a minimum aligned sequence length of 300 bp. We pre-selected contigs alignments if at least five regions of the same contig were aligned successively to the QTL region in a span shorter than 20 kb and covering at least 3% of the QTL sequence. Contigs were considered as valid if the alignment covered at least 40% of the contig length and 20% of the QTL length. For annotation, a *de novo* transposable elements (TEs) library was generated from the long reads with RepeatModeler2 (version 2.0.1, options -engine ncbi; *Flynn et al., 2020*). TEs were detected and removed using RepeatMasker (version 4.1.0; *Tarailo-Graovac and Chen, 2009*) with the *de novo* TEs library. Annotation of the genome was performed with MAKER2 (version 2.31.9; *Holt and Yandell, 2011*) using all ESTs sequences downloaded from the NCBI (organisms: *P. glaucum*, *Oryza sativa*, *Zea mays*, *Sorghum bicolor*, *P. miliaceum*, and *Setaria italica*), all protein sequences of *P. glaucum* available on the NCBI (July 2020), all protein sequences of annotated genes on the reference genome (http://gigadb.org/dataset/100192) and the Uniprot-Swiss-Prot protein database (section viridiplantae, release-2020_06). RNAseq data mapping was performed with TOGGLe v3 (*Tranchant-Dubreuil et al., 2018*). Reads were aligned to the long-read sequences with HISAT2 version 2.0.1 (*Kim et al., 2015*) and an annotation file was produced with Stringtie version 1.3.4 (*Pertea et al., 2015*). This file was used as input of MAKER2 (default parameters, the SNAP HMM *O. sativa* file model). Predicted protein sequences were aligned to the Uniprot-Swiss-Prot protein database (section viridiplantae) using

blastx (*Altschul et al., 1990*). Genes with protein domain signatures were recovered using Pfam database and InterProScan version 5.19-58.0 (-appl pfam -dp -f TSV -goterms -iprlookup -pa -t p; *Mulder and Apweiler, 2007*).

### *In situ* hybridization

Pearl millet root tip fixation, embedding, and sectioning step were performed as described in *Mounier et al., 2020*. Briefly, 7-day-old millet primary root tip (1.5 cm) were hand-dissected and the tip were immediately aligned between two biopsy foam (M476-1, Simport, Canada), and transferred in a cassette (1,267,796 Thermo Scientific, USA) in a 3:1 ethanol:acetic acid fixative solution. The samples were then placed in a fresh fixative solution and a 5-min vacuum was applied two times. The solution was then replaced by a fresh fixative solution and samples were stored over-night at 4°C. The samples were then subjected to several 5 min baths with increasing ethanol concentrations (75, 80, 85, 90, 95, and 100%), one bath in an ethanol/butanol (1:1) solution and one bath in absolute butanol on ice. The samples were transferred to a water bath at 54°C inside of a histology microwave oven (Histos 5 Rapid Tissue Processor, Milestone, Italy). The samples were then subjected to a bath in butanol/paraffin (1:1) solution at 54°C and 300 W and then two baths in paraffin at 54°C and 250 W. Prior to the embedding step, the root bundles were rapidly removed from the cassettes while the paraffin was still liquid and transferred to a cold RNase-free surface. The bundles were subsequently transferred vertically and placed upside down in a molding tray (E70182, EMS, USA). The paraffin blocks were maintained at 4°C and protected from light. Transversal sections with a thickness of 8 µm were cut on an RNase-free microtome (RM2255, Leica, Germany) and mounted on a cut edge frosted glass slide (VWR, 631-1551) prewarmed at 45°C with drops of RNase-free water. Slides were then baked at 60°C until the section was well fixed on the slide (around 20 min) prior to RNAscope.

The RNAscope assay was performed by Be More Specific. Probes against *PgGRXC9* were custom designed by ACD and are available in the ACD catalogue as Came-GRXC9-C1 (Ref. 1187801-C1). The probes were designed based on the *PgGRXC9* re-annotated sequence and were complementary to nucleotides at positions 71–671, excluding the 398–510 positions. The negative probe was provided by ACD (Accession # EF191515). The *in situ* hybridization assay was performed using the ACD RNAscope 2.5 HD Detection Reagent-RED kit (cat. no. 322360) using the provided protocol (http://www.acdbio.com/technical-support/user-manuals). Imaging of tissue sections was performed using a Nikon Eclipse NI-E (Nikon Corporation, Tokyo, Japan) with a 40X Plan Apo $\lambda$ 0.95 NA objectives. Images were processed using imageJ contrast enhancement tool to 0.001% of saturated pixel.

### Statistical methods

Statistical analyses were performed using R version 4.0.2 5 (*R Development Core Team, 2018*). Principal component analyses were performed using the prcomp() function. Pearson's correlation analyses were performed using the corr() function within the ggcorrplot package. The variance of each variable was partitioned into components attributable to the genotypic (line) and year in interaction with block as additional factor using an ANOVA (aov() function in the agricolae package). Adjusted means of the variables for the different lines across the two experiments were further calculated using the least-squares means lsmeans() function (lsmeans package).

## Acknowledgements

We dedicate this article to the memory of our colleague Yann Guédon (CIRAD). This work was supported by the French Institute for Sustainable Development (IRD), the French Ministry for Research and Higher Education (PhD grant to SP), the Agence National pour la Recherche (RootAdapt grant n° ANR-17-CE20-0022 to LL), the Agropolis Fondation (NewPearl grant n° AF 1301-015 in the frame of the CERES initiative and ValoRoot grant n°2202-002 to LL and Flagship Project CalClim grant no. 1802-002 to JPR) as part of the 'Investissement d'avenir' (ANR-l0-LABX-0001-0l), under the frame of I-SITE MUSE (ANR-16-IDEX-0006), by the Fondazione Cariplo (n° FC 2013-0891) and by the CGIAR Research Programme on Grain Legumes and Dryland Cereals (GLDC). Financial support by the Access to Research Infrastructures activity in the Horizon 2020 Programme of the EU (EPPN2020 Grant Agreement 731013) is gratefully acknowledged. The authors acknowledge support from the iTrop High-performance computing platform (member of the South Green Platform) at IRD Montpellier.

# Additional information

## Competing interests

Thibault Mounier: is affiliated with Be More Specific. The author has no financial interests to declare. The other authors declare that no competing interests exist.

## Funding

| Funder | Grant reference number | Author |
|---|---|---|
| Institut de Recherche pour le Développement | | Laurent Laplaze |
| French Ministry for Research and Higher Education | PhD grant | Sixtine Passot |
| Agence Nationale de la Recherche | ANR-17-CE20-0022 | Laurent Laplaze |
| Agropolis Fondation | AF 1301-015 | Laurent Laplaze |
| Agropolis Fondation | 2202-002 | Laurent Laplaze |
| Agropolis Fondation | 1802-002 | Jean-Philippe Reichheld |
| Agence Nationale de la Recherche | ANR-l0-LABX-0001-0l | Laurent Laplaze |
| Agence Nationale de la Recherche | ANR-16-IDEX-0006 | Laurent Laplaze |
| Horizon 2020 Framework Programme | 731013 | Laurent Laplaze |

The funders had no role in study design, data collection, and interpretation, or the decision to submit the work for publication.

## Author contributions

Carla de la Fuente, Jonathan A Atkinson, Sixtine Passot, Data curation, Formal analysis, Investigation, Writing – original draft, Writing – review and editing; Alexandre Grondin, Formal analysis, Investigation, Writing – original draft, Writing – review and editing; Bassirou Sine, Christophe Belin, Thibault Mounier, Formal analysis, Investigation, Writing – review and editing; Marilyne Debieu, Marine Salson, Julie Orjuela, Christine Tranchant-Dubreuil, Jean-Rémy Brossier, Hang Ngan Dinh, Antony Champion, Anne-Sophie Petitot, Celia Barrachina, Marine Pratlong, Yann Guedon, Vincent Vadez, Jean-Philippe Reichheld, Data curation, Formal analysis, Investigation, Writing – review and editing; Amir Hajjarpoor, Conceptualization, Formal analysis, Supervision, Funding acquisition, Writing – original draft, Project administration, Writing – review and editing; Maxime Steffen, Data curation, Investigation, Writing – review and editing; Charlotte Morgado, Data curation, Formal analysis, Investigation; Bipin K Pandey, Julie Darmau, Data curation, Formal analysis, Investigation, Methodology, Writing – review and editing; Princia Nakombo-Gbassault, Formal analysis; Pascal Gantet, Prakash Gangashetty, Resources, Formal analysis, Investigation, Writing – review and editing; Malcolm J Bennett, Resources, Data curation, Formal analysis, Investigation, Writing – review and editing; Ndjido Ardo Kane, Resources, Data curation, Formal analysis, Investigation, Methodology, Writing – review and editing; Soazig Guyomarc'h, Formal analysis, Investigation, Methodology, Writing – review and editing; Darren M Wells, Conceptualization, Data curation, Formal analysis, Supervision, Investigation, Writing – review and editing; Yves Vigouroux, Conceptualization, Data curation, Formal analysis, Supervision, Funding acquisition, Investigation, Writing – original draft, Project administration, Writing – review and editing; Laurent Laplaze, Conceptualization, Formal analysis, Supervision, Funding acquisition, Investigation, Writing – original draft, Project administration, Writing – review and editing

## Author ORCIDs

Alexandre Grondin http://orcid.org/0000-0001-6726-6274
Christophe Belin http://orcid.org/0000-0003-2129-5349
Julie Orjuela http://orcid.org/0000-0001-8387-2266

Bipin K Pandey ⓘ https://orcid.org/0000-0002-9614-1347
Pascal Gantet ⓘ http://orcid.org/0000-0003-1314-0187
Ndjido Ardo Kane ⓘ http://orcid.org/0000-0002-1879-019X
Darren M Wells ⓘ http://orcid.org/0000-0002-4246-4909
Yves Vigouroux ⓘ http://orcid.org/0000-0002-8361-6040
Laurent Laplaze ⓘ http://orcid.org/0000-0002-6568-6504

Reviewer #1 (Public Review): https://doi.org/10.7554/eLife.86169.3.sa1
Reviewer #2 (Public Review): https://doi.org/10.7554/eLife.86169.3.sa2
Author Response https://doi.org/10.7554/eLife.86169.3.sa3

---

## Additional files

### Supplementary files
- MDAR checklist
- Supplementary file 1. Supplementary tables (Tables S1 to S5).

### Data availability
Genomic data are available at the National Center for Biotechnology Information (NCBI) under reference number GCA_002174835.2. Genotyping (GBS) data are available in NCBI BioProject under reference number PRJNA492967 (GWAS) and PRJNA769524 (BSA). RNAseq data are available in the Gene Expression Omnibus (GEO) under reference GSE185517. Oxford Nanopore long reads are available on the European Nucleotide Archive (ENA) under reference ERR12178246 and ERR10627707. Field trials data are available at Dryad under reference https://doi.org/10.5061/dryad.qv9s4mwk2.

The following datasets were generated:

| Author(s) | Year | Dataset title | Dataset URL | Database and Identifier |
|---|---|---|---|---|
| Laplaze L, Debieu M, Grondin A | 2021 | Comparison of the root tip transcriptome of 2 pearl millet lines with contrasted primary root growth | https://www.ncbi.nlm.nih.gov/geo/query/acc.cgi?acc=GSE185517 | NCBI Gene Expression Omnibus, GSE185517 |
| Vigouroux Y | 2018 | Pearl millet association mapping | https://www.ncbi.nlm.nih.gov/bioproject/?term=PRJNA492967 | NCBI BioProject, PRJNA492967 |
| Vigouroux Y | 2021 | Bulk segregant analysis of pearl millet root traits | https://www.ncbi.nlm.nih.gov/bioproject/?term=PRJNA769524 | NCBI BioProject, PRJNA769524 |
| Vigouroux Y | 2023 | PromethION sequencing; Oxford Nanopore long reads from Cenchrus americanus | https://www.ebi.ac.uk/ena/browser/view/ERR10627707 | EBI European Nucleotide Archive, ERR10627707 |
| Grondin A, Laplaze L | 2023 | Data from: Glutaredoxin regulation of primary root growth confers early drought stress tolerance in pearl millet | https://doi.org/10.5061/dryad.qv9s4mwk2 | Dryad Digital Repository, 10.5061/dryad.qv9s4mwk2 |
| Vigouroux Y | 2023 | MinION sequencing; Oxford Nanopore long reads from Cenchrus americanus Pod103sr8 genotype | https://www.ebi.ac.uk/ena/browser/view/ERR12178246 | EBI European Nucleotide Archive, ERR12178246 |

The following previously published dataset was used:

| Author(s) | Year | Dataset title | Dataset URL | Database and Identifier |
|---|---|---|---|---|
| Pearl Millet | 2018 | Genome assembly ASM217483v2 | https://www.ncbi.nlm.nih.gov/datasets/genome/GCA_002174835.2/ | NCBI Assembly, GCA_002174835.2 |

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
