## [Editor Report · eLife assessment]

This is an **important** paper that combines methods ranging from agronomy and plant breeding to *Arabidopsis* functional genetics, to argue that polymorphism in a single gene affects crop yield in pearl millet by affecting root cell elongation and drought stress resilience in a poorly studied crop. The overall argument is plausible but whether the **solid** evidence generated with *Arabidopsis* experiments can be extended to pearl millet itself is unclear.

---

## [Referee Report · Reviewer #1 (Public Review)]

The authors use a combination of crop modeling and field experiments to argue that drought during seedling establishment likely severely impacts the yield of pearl millet, an important but understudied cereal crop and that rapid seedling root elongation could play a major role in mitigating this. They further argue that this trait has a strong genetic basis and that major polymorphisms in candidate genes can be identified using standard methods from modern genetics and genomics. Finally, they use homology with the model plant *Arabidopsis thaliana* to argue that the function of one putatively causal gene is to regulate root cell elongation.

The major strength of this paper is that it convincingly demonstrates how modern methods from plant breeding and model organisms can be combined to address questions of great practical importance in important but poorly understood crops. The notion that it is possible to connect single-locus polymorphism and cellular biology to drought tolerance and crop yield in pearl millet is not a trivial one.

The weakness is obvious: while the argument made is convincing, it must be recognized that the strength of the evidence is by no means of the level expected in a model organism. Conclusions could easily be wrong, and there is no direct evidence that regulatory variation in PgGRXC9 leads to higher crop yield via cell elongation and seedling drought tolerance. However, generating such evidence in a poorly studied crop would be a monumental undertaking, and should probably not be the priority of people working on pearl millet!

The utility of this work is that it suggests that it is practicable to gain valuable insight into crop adaptation by clever use of modern methods from a variety of sources.

---

## [Referee Report · Reviewer #2 (Public Review)]

Carla de la Fuente et al., utilize a diversity of approaches to understand which plant traits contribute to the stress resilience of pearl millet in the Sahelian desert environment. By comparing data resulting from crop modeling of pearl millet growth and meteorological data from a span of 20 years, the authors clearly determined that early season drought resilience is contributed by accelerated growth of the seedling primary root, which confirms a hypothesis generated in a previous study, Passot et al., 2016. To determine the genetic basis for this trait, they performed a combination of GWAS, QTL analysis, and RNA sequencing and identified a previously unannotated coding sequence of a glutaredoxin C9-like protein, PgGRXC9, as the strongest candidate. Phenotypic analysis using a mutant of the closest *Arabidopsis* homolog AtROXY19 suggests the broad conservation of this pathway. Comparisons between the transcript of PgGRXC9 by in situ hybridization (this work) and AtROXY19 pattern expression (Belin et al., 2014) support the hypothesis that this pathway acts in the elongation zone of the root. Additional analysis of cell production and elongation rates in root apex in both pearl millet and *A. thaliana* suggests that PgGRXC9 specifically regulates primary root through the promotion of cell elongation. While several studies have established the connection between redox status of cells and root growth, the current study represents an important contribution to the field because of the agricultural importance of the plant studied, and the connection made between this developmental trait and stress resilience in a specific and stressful environmental context of the Sahelian desert.

---

## [Author Response]

The following is the authors’ response to the original reviews.

**Reviewer #1 (Recommendations For The Authors):**
Some suggestions:1. It's obviously concerning that your GWAS results are not at all robust to the approach used (Fig S3). Did you try something non-parametric, like a Kruskal-Wallis test?

We used both GWAS and crosses (F2) to validate the presence of the QTL. So ,evidence is not only brought by GWAS. We did not use non parametric tests as we will have difficulty to account for population structure/relatedness with such approaches. Our GWAS approach is certainly a little underpowered associated with the number of individuals we used and certainly the polygenic nature of the root growth traits. But F2 crosses allow us to put more evidence weight on some region we identified with GWAS.

1. You don't explain what you do with heterozygotes, nor discuss the level of inbreeding in general.

We are dealing with inbred lines, but indeed there are not completely fixed inbred lines. For the remaining heterozygotes, they were randomly fixed in one or the other alleles. The median heterozygosity value was low at 5.6%. We clarified this point in the material and methods.

1. The finding that over 30% of RNA-seq reads don't seem to have an annotated home should give you pause. Do they map anywhere? At least discuss what is going on. Also, note that you likely have enormous errors in SNP-calling due to cryptic structural variation - think about what this might do?

We agree with reviewer #1. We added a few sentences in the result section to clarify this point: “When further analyzed, 15.15% of the unmapped reads (with no correspondence to predicted CDS) were found not to match the reference genome. These might correspond either to unsequenced regions or to genotype-specific genomic regions that are not present in the reference line. The remaining unmapped reads corresponded to either rRNA and tRNA genes (40.28% of the unmapped reads) or to non-annotated genes or non-coding RNAs (44.57% of the unmapped reads).”As we used the same reference genome for mapping the RNAseq reads, some genes might not being present in our analysis for the two lines we studied.

1. Did you consider moving PgGRXC9 into *Arabidopsis*?

This is a great suggestion. In fact, we plan to explore more how some GRXs regulate root growth and how this is conserved in plants in a follow up project. This is however beyond the scope of this manuscript.

Minor suggestions:1. Why not calculate H^2 simply as line variance divided by total?

Heritability estimated on single individuals in population, approaches generally used for human and animal breeding led directly to line variance divided by total phenotypic variance.

But in plant breeding (or plant science), we generally work on replicated genotypes in different blocks/experimental repetition. So we estimate the heritability of the mean phenotype of genotypes. There is ample literature (Nyquist, 1991; Holland et al. 2003; for a very nice and smartly written explanation, on the introduction of this PhD: http://opus.uni-hohenheim.de/volltexte/2020/1720/pdf/20200221_PhD_Thesis_Publikationsversion.pdf). Calculation of heritability (of the mean phenotype) should take into account for the calculation of the phenotypic variance (denominator) the number of replicate genotypes (we do not have a single plant, but several clones when using inbred lines: n). The meaning of the formula is that the error in the model is inflated because we have n replicate plants per genotype. And so to estimate the heritability of the average genotype, we have to take into account this inflated variance in the errors.

1. While the paper overall is well-written, the captions need further proof-reading.

We corrected all the captions.

**Reviewer #2 (Recommendations For The Authors):**
Major suggestions:1. The experimental support for the mutant phenotype of roxy19 needs to be further substantiated. Current methods available for CRISPR mutagenesis make it relatively easy to generate additional alleles. Alternatively, the authors could complement the mutant with a wild-type copy of the gene. These approaches represent the standard of the field and should be used here as well.

We agree with rev #2. We added some sentences in the discussion to stress out the limitations of our study to link the QTL to PgGRXC9.

As stated above we’d like to explore more how some GRXs regulate root growth and how this is conserved in plants. We plan to generate new single and multiple mutants in ROXY19 and its closest homologues (using CRISPR). This is, however, beyond this manuscript.

1. The authors may want to state more clearly what the hypothesis is for how redox levels might contribute to root length differences and more clearly state what the limits of their current study are.

We modified the discussion to try to clearly indicate the limitations of our study.

1. Differences in root growth can be the consequence of a number of different parameters that contribute to root elongation and the authors need to more clearly define which of these are likely affected in their different genotypes.

We agree with Reviewer #2. However, as stated before, we plan to further explore the molecular and cellular mechanisms responsible for the phenotype we observe in *Arabidopsis*. This will need extra work and is beyond the scope of this manuscript.

1. Page 13, first paragraph. The authors provide an overly strong statement that suggests they have determined the molecular basis for the difference in PgGRXC9:" Altogether, our results suggest that PgGRXC9 is a positive regulator of root growth and that a polymorphism in the promoter region of PgGRXC9 associated with changes in its expression level appeared responsible for a quantitative difference in root growth between the two lines."While their results suggest the PgGRXC9 locus is associated with root growth variation, they have not directly tested the effect of the polymorphisms in the promoter on gene expression and this statement needs to be weakened.

We changed the text to: “Altogether, our results suggest that PgGRXC9 is a positive regulator of root growth and that a polymorphism in the promoter region of PgGRXC9 might led to changes in its expression level and ultimately to a quantitative difference in root growth between the two lines. However, the effect of the polymorphisms in the promoter on gene expression need to be tested to validate this hypothesis.”

We also changed the title of the manuscript to better reflect our results.

Minor suggestions:1. Page 4: "FTSW below 0.3 was considered a stressful condition." It was not specified how this threshold was determined.

This value corresponds to the measured FTSW value at which pearl millet genotypes subjected to a dry down generally start to reduce their transpiration rate (see Fig. 1 of Kholová et al, 2010; https://doi.org/10.1093/jxb/erp314). At FTSW values above 0.3, transpiration is not affected. At FTSW values around 0.3, the water supply from pearl millet roots cannot fully support transpiration. The plant enters a drought stress responsive phase and progressively closes its stomata to reduce water losses and decrease plant productive functions to match water supply. We have clarified this in the manuscript.

1. Page 6: Figure 1; footnote: at the end of the description of panel A, a comma is missing between "red" and "blue."

Thanks for pointing that out. This was corrected.

1. The root growth data determined by X-ray imaging is not significant (Fig S4B), yet the authors describe the result in the main text without qualification. The authors should clarify this in the text.

We added some text to clarify this.

1. Page 9: Figure 2C; It would be better to enlarge these images and annotate them to indicate what specific anatomical features have been measured. Currently, only an expert in the field would be able to interpret these images.

While we understand the point made by Reviewer #2, Fig2C was meant to illustrate differences in the root tip of the two lines.

1. Page 9: Figures 2D and E; the number of biological samples measured is not indicated (what is "n"?).

Thanks again for pointing this out. This was added to the figure legend.

1. Page 14: Figure 4B; scale bar needs to be included.

Scale bars were added to the pictures.

1. Page 14: Figure 4; I recommend adding confocal images or DIC of cleared root apex tissues to easily compare the RAM size and cell lengths in both WT and roxy19 mutant.

Once again, we plan to have a follow up study on the molecular and cellular mechanisms of action of ROXY19 and its closest homologues on root development. We believe a thorough analysis of differences in phenotype could be illustrated in a future manuscript.

1. Page 18: main text; "we propose that redox regulation in the root meristem is responsible for a root growth QTL in pearl millet." This statement is ambiguous in the description of the mechanism. The authors do not clarify if the role they propose for PgGRXC9 is in the meristematic or elongation zone. Likely the authors are not able to know precisely where the gene is acting at this point, and so the presented hypothesis needs to more clearly state what limitations there are in assigning a mode of action for the PgGRXC9 and ROXY19 genes in root growth.

We rewrote this paragraph to clarify the current gap in our understanding of the putative PgGRXC9 function.